



# Statistical summaries for streamed data from climate simulations: One-pass algorithms (v0.6.2)

Katherine Grayson[1], Stephan Thober[2], Aleksander Lacima-Nadolnik[1], Ehsan Sharifi[2], Llorenç Lledó[4], and Francisco Doblas-Reyes[1,3]

[1]Earth Sciences Department, Barcelona Supercomputing Center, Barcelona, 08034, Spain
[2]Department of Computational Hydrosystems, Helmholtz Centre for Environmental Research, Leipzig, 04318, Germany
[3]Institució Catalana de Recerca i Estudis Avançats, Barcelona, 08010, Spain
[4]ECMWF, Bonn, 53177, Germany

**Correspondence:**
Katherine Grayson (katherine.grayson@bsc.es)

**Abstract.** Projections from global climate models (GCMs) are a fundamental information source for climate adaptation policies and socio-economic decisions. As such, these models are being progressively run at finer spatio-temporal resolutions to resolve smaller scale dynamics and consequently reduce uncertainty associated with parameterizations. Yet even with increased capacity from High Performance Computing (HPC) the consequent size of the data output (which can be on the order of Terabytes to Petabytes), means that native resolution data cannot feasibly be stored for long time periods. Lower resolution archives containing a reduced set of variables are often all that is kept, limiting data consumers from harnessing the full potential of these models. To overcome this growing challenge, the climate modelling community is investigating data streaming; a novel way of processing GCM output without having to store a limited set of variables on disk. In this paper we present a detailed analysis of the use of one-pass algorithms from the 'one-pass' package, for streamed climate data. These intelligent data reduction techniques allow for the computation of statistics on-the-fly, enabling climate workflows to temporally aggregate the data output from GCMs into meaningful statistics for the end-user without having to store the full time series. We present these algorithms for four different statistics: mean, standard deviation, percentiles and histograms. Each statistic is presented in the context of a use case, showing the statistic applied to a relevant variable. For statistics that can be represented by a single floating point value (i.e., mean, standard deviation, variance), the accuracy is at the order of the numerical precision of the machine and the memory savings scale linearly with the period of time covered by the statistic. For the statistics that require a distribution (percentiles and histograms), we present an algorithm that reduces the full time series to a set of key clusters that represent the distribution. Using this algorithm we find that the accuracy provided is well within the acceptable bounds for the climate variables examined while still providing memory savings that bypass the unfeasible storage requirements of high-resolution data.





## 1 Introduction

Understanding how extreme events, both at regional and global scales, are going to impact society under climate change is of ever growing importance (Seneviratne et al., 2012; Rising et al., 2022; Pörtner et al., 2022). Projections from global climate models (GCMs) are regularly used to create information for climate adaptation policies and socio-economic decisions. As demand for accuracy in these projections grows, GCMs are being run at increasingly finer spatio-temporal resolution to capture

both small-scale processes, such as convective storms and oceanic eddies, as well as larger atmospheric dynamics through the improved representation of the interaction between small and large scale dynamical processes (Palmer, 2014; Bador et al., 2020; Iles et al., 2020; Stevens et al., 2020; Rackow et al., 2024). These increases in resolution are also beneficial for end-user analyses or applications, that typically require local information and frequent time samples to produce accurate estimates of climate impacts (Katopodis et al., 2021; Orr et al., 2021).

Due to the ongoing movement in the climate community towards increasingly higher spatio-temporal resolutions of climate data, questions are beginning to arise as to how this data will be managed (Hu et al., 2018; Bauer et al., 2021). The growing size of the data output makes the current state-of-the-art archives (e.g., Coordinated Regional Climate Downscaling Experiment (CORDEX) and Coupled Model Intercomparison Project (CMIP)), unfeasible. Moreover, the current archival method has left some data users without their required data as climate model protocols either limit the number of variables stored or reduce their

resolution and frequency (e.g., by storing monthly means or interpolated grids) to cope with the size of the archives. As such, initiatives such as Destination Earth (DestinE) (Bauer et al., 2021; Hoffmann et al., 2023) are investigating the novel method of data streaming, where output arrives to the data consumer in a continuous stream (Kolajo et al., 2019; Marinescu, 2023). Data streaming allows access to the climate data at the highest frequency available (e.g., hourly), at native spatial resolution in near-real model run-time. In the context of the climate impact community, this means that climate impact information can be created

alongside the climate model, without having to wait for the full simulation to be completed. This provides an unprecedented time-scale reduction to access the data and produce meaningful output compared with the current simulation paradigm (e.g., CMIP) and the possibility of using variables and frequencies not previously available.

Yet the advent of data streaming in the climate community poses its own set of challenges. Often, downstream data users require climate data that spans long temporal periods. For example, many hydrological impact models require daily, monthly

or annual maximum precipitation values (Teutschbein and Seibert, 2012; Samaniego et al., 2019), while in the wind energy sector, accurate distribution functions of the wind speed are essential (Pryor and Barthelmie, 2010; Lledo, 2019). Obtaining these statistics that span time scales that are potentially longer than the data streaming window can no longer be done using the traditional methods. As data in a stream can only be accessed once, this introduces the one-pass problem; how to compute summaries, diagnostics or derived quantities without having access to the whole time series?

In this paper we present a detailed analysis of the use of one-pass algorithms and how they work as intelligent data reduction techniques for streamed climate simulation data. Normally, the computation of statistics is done using the conventional method, where two sequential passes are made through a dataset, first to gather relevant information, and then to perform calculations based on that information. This method requires having the entire dataset available when the computation is performed and we





will refer to it as the 'conventional' method throughout the paper to signify the common way of calculating statistics. Unlike the conventional method, one-pass algorithms do not have access to a whole time series, rather, they process data incrementally every time that the model outputs new time steps (Muthukrishnan, 2005). This is done by sequentially processing data chunks as they become available, with each chunk's value being incorporated into a rolling summary which is then moved into an output buffer before processing the next chunk. While these algorithms have been adopted in other fields such as online trading (Loveless et al., 2013) and machine learning (Min et al., 2022), they have yet to find a foothold in climate science, mainly because they have not been necessary until now. Through this work, we show the foundations of the infrastructure required for the climate community to harness the capabilities of high spatio-temporal climate data through data streaming.

This paper is organised as follows. In Sect. 2 we present the mathematical notation used throughout this paper to describe the statistics. Sects. 3 to 5 then cover the algorithms used for the mean, standard deviation and distributions respectively. These statistics have been chosen as they represent the most commonly required statistics for climate data, however many other statistics (i.e., minimum, maximum, threshold exceedance etc.) can be implemented using the same approach. For each statistic, the one-pass algorithm is first presented, followed by a use case example which applies the algorithm to a relevant variable over a meaningful time span. With the aid of these use cases, the numerical accuracy compared to the conventional approach (being able to read the dataset as a whole to compute the statistic) are given, along with the memory savings provided. In Sect. 6 we discuss the concept of convergence and how the one-pass statistics can be used for bias-adjustment of streamed climate data. Finally conclusions are drawn in 7. We further note that the full Python implementation of these algorithms, ready for use in a streaming workflow, can be found at Grayson (2025) (v0.6.2) and request the reader to follow the documentation for details on implementation.

## 2 Mathematical notation

For a given dataset, the following mathematical notation is used to describe the one-pass algorithms:

- $n$ is the current number of data samples (time steps) passed to the statistic.

- $w$ is the length of the incoming data chunk (number of time steps).

- $c$ is the number of time steps required to complete the statistic (i.e., if the model provides hourly output and we require a daily statistic, $c = 24$).

- $x_n$ is one time step of the data at time $t = n$.

- $X_n = \{x_1, x_2, \ldots, x_n\}$ represents the full dataset up to $t = n$.

- $X_w = \{x_{n+1}, \ldots, x_{n+w}\}$, is the incoming data chunk of length $w$.

- $S_n$ is the rolling summary of the statistic before the new chunk at time $t = n$. This summary varies for each statistic i.e., if it is the mean statistic $S_n = \bar{X}_n$. This rolling summary will always be of length one in the time dimension.





   – $g$ is a one-pass function that updates the previous summary $S_n$ with then new incoming data $X_w$.

We introduce the chunk length $w$ as, in many cases, a data stream containing a few consecutive time steps will be outputted from a climate model in one go. In the case where the incoming data stream has only one time step, $(w = 1)$, $X_w$, reduces to $x_{n+1}$.

## 3   Mean

### 3.1   Algorithm description

The one-pass algorithm for the mean is given by

$$\bar{X}_{n+w} = g(S_n, X_w) = \bar{X}_n + w\left(\frac{\bar{X}_w - \bar{X}_n}{n + w}\right), \tag{1}$$

where $\bar{X}_{n+w}$ is the updated rolling mean of the dataset with the new data chunk $X_w$ and the rolling summary $S_n$ is given by the rolling mean $\bar{X}_n$. If $w > 1, \bar{X}_w$, is the temporal conventional mean over the incoming data chunk, however if the data is streamed at the same frequency of the model output with $w = 1$ then $\bar{X}_w = x_{n+1}$, where $x_{n+1}$ is the incoming time stamp.

### 95  3.2   Temperature

We apply the mean one-pass algorithm given in eq. (1) in the context of temperature. Understanding the average trends of temperature is crucial, especially considering the ongoing global and regional occurrences of temperature extremes (Mikkonen et al., 2015; Russo et al., 2015). New European projects such as NextGEMS (a) and DestinationEarth (2024) are aiming to provide global climate projections for a variety of variables at spatial resolutions ranging from 0.025 to 0.1°. This will

allow for granular analysis of projected temperatures to inform climate adaptation at regional scale, yet performing basic computations with such vast amounts of data can prove challenging. In this use case for the mean algorithm, we use data from ECMWF's Integrated Forecasting System (IFS) model coupled with the Finite Element / Volume Sea-ice Ocean Model (FESOM) (experiment tco2559-ng5-cycle3) (Koldunov et al., 2023; Rackow et al., 2024) (run as part of the NextGEMS (a) project), looking at the 2 m temperature over March 2020. We use the hourly data at native spatial resolution ($\sim 0.04°$),

resulting in a global map containing approximately 26.31 million spatial grid cells, 744 time steps and a full size of 145.82 GB with double precision (float64).

     We calculated the March monthly mean of this data using both the conventional method and the one-pass algorithm given in eq. (1). Computing the temporal conventional mean of this data requires the full time series to be loaded into memory, summed across every cell then divided by the length of the time dimension (in this case 744). Due to the high memory requirements of

this dataset, this was performed on a high memory node (256 GB) on the Levante supercomputer. The data set was re-chunked into 10 spatial chunks using the Python library dask-xarray (Dask, 2024), where each chunk could fit into available memory. The conventional mean was then computed on each chunk. For the one-pass computation, we used a chunk length of $w = 1$ and called into memory each hour $x_n$ of the dataset to simulate streaming, iteratively updating eq. (1) until $n = c$. These two



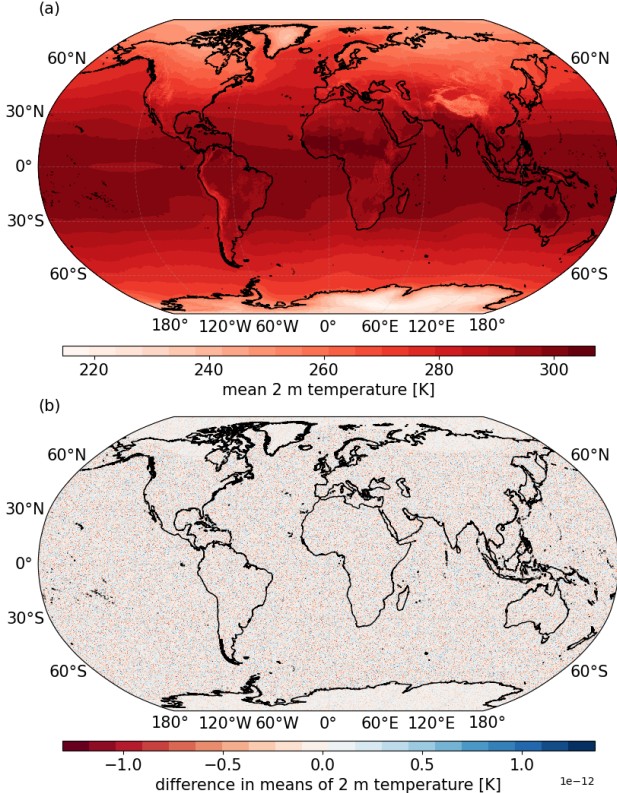

**Figure 1.** (a) Global monthly mean of the 2 m temperature over March 2020 using hourly data from the IFS model, computed using the one-pass algorithm method given in eq. (1). (b) The absolute difference between (a) and the mean calculated using the conventional method, calculated using the python package numpy (Harris et al., 2020).

methods highlight that with such large data sets the conventional mean is not necessarily the simplest approach, as we still

need specialised tools and high memory resources for computation. Rather than adding additional complexity, the one-pass approach allows for simpler handling and easier computation.

The results of the one-pass mean can be seen in Figure 1(a). We note that for plotting convenience the native grid was interpolated to a 0.1° regular lat-lon. Figure 1(b) shows the absolute difference between the conventional and the one-pass mean shown in (a). The difference is represented by randomly distributed noise at the order of $10^{-12}$, consistent with the

numerical precision of the data. This negligible absolute difference is therefore attributed to the machine precision as opposed to algorithmic discrepancies and is well below the accuracy required for the variable.

With regards to the memory savings, the one-pass method requires only two data memory blocks, $\bar{X}_w$ and $x_n$, both approximately 200 MB with double precision (which is the size of one global time step), to be kept in memory at any one point in time. If this computation was performed in real streaming mode this would require only ∼400 MB of memory to compute

the monthly mean, 2/744$^{\text{th}}$ of the 145.82 GB memory requirements for the conventional method. We also note here that the



memory cost of the one-pass is independent of the length of the statistic span. For the case of $w = 1$, the memory requirements will always be twice that of storing a spatial field, as opposed to the conventional method where the memory requirements will increase linearly with the length of the time series required to compute the statistic.

Moreover, this computation demonstrates that the one-pass do not merely provide memory savings; they provide user-oriented diagnostics that allow for easy computation. Due to this vast reduction in memory requirements, different variables can be computed in parallel to each other, further aiding usability. The current paradigm of loading the entire dataset is not practical (and in some cases not possible) for data of this magnitude. Indeed, special tools such as Python's dask-xarray packages are often required.

## 4 Standard deviation

### 4.1 Algorithm description

The one-pass algorithm for the standard deviation (and also variance) is calculated over the requested temporal frequency $c$, by updating two estimates iteratively; the one-pass mean and the sum of the squared differences. First, let the conventional summary for the sum of the squared differences, $M_c$, be defined as

$$M_c = \sum_{n=1}^{c} \left( x_n - \bar{X}_c \right)^2, \tag{2}$$

where $\bar{X}_c$ is the conventional mean of the whole dataset $X_c$ required for the statistic. For the one-pass calculation of the standard deviation the rolling summary $S_n$ defines the sum of the squared differences, $M_n$. In the case where $w = 1$, the rolling summary is updated by

$$M_{n+1} = g(S_n, x_{n+1})$$
$$= M_n + \left( x_{n+1} - \bar{X}_n \right)\left( x_{n+1} - \bar{X}_{n+1} \right), \tag{3}$$

where $\bar{X}_n$ and $\bar{X}_{n+1}$ are both given by the algorithm for the one-pass mean in eq. (1). Eq. (3) is known as Welford's algorithm. In the case where the incoming data has more than one time step ($w > 1$), $M_n$ is updated by

$$M_{n+w} = g(S_n, X_w)$$
$$= M_n + M_w + \frac{wn\left( \bar{X}_n - \bar{X}_w \right)^2}{n + w}, \tag{4}$$

where $M_w$ is the conventional sum of the squared differences over the incoming data block of length $w$ (given by eq. (2) with $c = w$), $\bar{X}_n$ is the one-pass mean at $t = n$ calculated with eq. (1) and $\bar{X}_w$ is the conventional mean of the incoming data block. See Mastelini and de Carvalho (2021) for details.

Once enough data has been added to the rolling summary $M_n$ so that $n = c$ we calculate the standard deviation $\sigma$ using

$$\sigma = \sqrt{\frac{M_n}{n-1}}, \tag{5}$$

where $M_n$ is divided by $n - 1$ to obtain the sample variance. Eq. (5) also applies to the conventional summary $M_c$.





## 4.2 Sea surface height variability

We apply the one-pass algorithm for standard deviation to the sea surface height (ssh). When evaluating the output of any
climate model it is necessary to check scientific soundness and quantify uncertainty through quality assessment checks. For
ocean data, one such method of soundness can be the standard deviation of the ssh between different model ensembles com-
pared to satellite altimetry data. The ssh can be used to better understand ocean dynamics as its variability gives insights into
the redistribution of mass, heat and salt within the water column (Close et al., 2020).

We calculate the annual standard deviation using data from the FESOM model (experiment tco2559-ng5-cycle3) (Rackow
et al., 2024), again run as part of the NextGEMS (a) project. We use daily data over 2021 at native resolution ($\sim$0.05°), making
an annual time series - comprised of 7.4 million grid cells and 365 time slices - of 10.09 GB using single precision (float32).
We first calculate the standard deviation using the conventional method defined in eq. (2), implemented with Python's numpy
package (Harris et al., 2020), followed by the one-pass method in eq. (4), with $w = 2$. Like with the mean calculation, for the
conventional calculation the data was spatially rechunked into 10 chunks using the Python library dask-xarray and each chunk
was computed separately, while for the one-pass, two daily time steps were iteratively called into memory to simulate data
streaming.

Figure 2(a) shows the one-pass standard deviation calculated using eq. (4) and eq. (5). Figure 2(b) then shows the difference
between the one-pass and the conventional calculation. Again, for plotting convenience the native grid was interpolated to a
0.25° regular lat-lon. Here the order of magnitude on the difference is $10^{-16}$; even smaller compared with the mean difference
in Fig. 1(b). Interestingly we also see some structure emerging in the differences shown in Fig. 2(b) which correlates with areas
of larger standard deviation, however due to the extremely values it is considered negligible in comparison to the required
accuracy of the statistic. Therefore, as with the mean statistic, this difference can also be attributed to machine precision
limitations.

The memory savings for the standard deviation are slightly less than the mean one-pass algorithm as here, in the case of
$w > 1$, other than the current data memory block $X_w$, four additional data summaries are required to be kept in memory, $M_n$,
$M_w$, $\bar{X}_n$, $\bar{X}_w$. Yet, as with the mean, these memory requirements are independent of the time-span (sample size) of the statistic
and do not increase as the number of values required to complete the statistic ($c$) increases. In the example presented here with
$w = 2$, approximately 112 MB (single precision) is required in memory, as opposed to the full 10.09 GB of the full dataset.
This is a reduction of two orders of magnitude for the memory requirements.

## 5 Distributions: percentiles and histograms

### 5.1 Algorithm description

Unlike the one-pass algorithms for mean and standard deviation (and others such as minimum, maximum, threshold ex-
ceedance), where the rolling summary $S_n$ can be described by one floating point value, estimates of a distribution cannot
be condensed in such a way. The t-digest algorithm has been developed by Dunning and Ertl (2019) and Dunning (2021) to





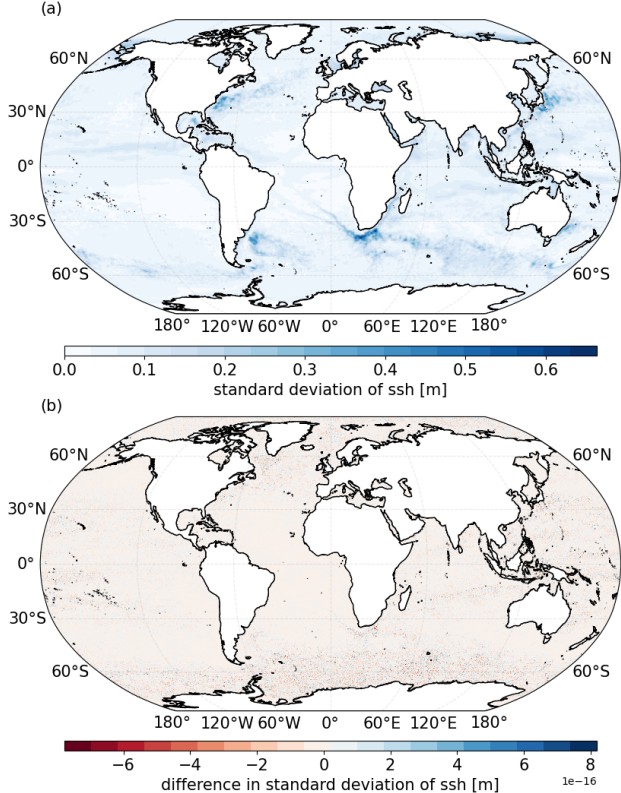

**Figure 2.** (a) Global annual standard deviation of the sea surface height over 2021 using daily data from the FESOM model, computed using the one-pass method given in eq. (4) and eq. (5). (b) The difference between the one-pass computation and the conventional computation.

create reliable estimates of probability distribution functions with one pass through the dataset. The t-digest algorithm is used here - to the best of our knowledge - for the first time on climate data.

   The t-digest algorithm is a clustering algorithm where the dataset, $X_n$, is represented by a series of clusters. Each cluster is summarised by a mean value and a corresponding weight, representing the average value in the cluster and the number of samples that have contributed to the cluster respectively. The data is added to each cluster in such a way that clusters

corresponding to the extremes of the distribution will contain far fewer samples than those around the median quantile, meaning that the error is relative to the quantile, as opposed to a constant absolute error seen in previous methods (Dunning and Ertl, 2019).

   For all the results presented in Sect. 5 we use the Python package crick (Crist-Harif, 2023) for the implementation of the t-digest algorithm and note that other packages may provide different results. There are two different algorithmic methods

within t-digest, one known as merging and one as clustering. Here we focus on the clustering algorithm, which adds each value in a streamed data chunk, $X_w$, to its nearest cluster. The unequal size (i.e., number of samples) contributing these clusters are set by the scale function. While there are different scale functions that can be used, the implementation of the t-digest used in



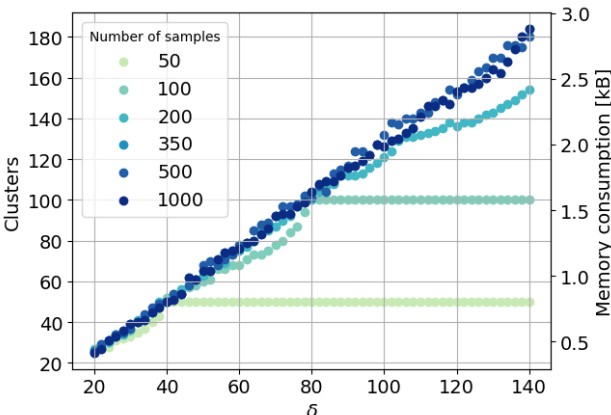

**Figure 3.** Number of clusters used to represent datasets of varying lengths as a function of the compression parameter $\delta$. The corresponding memory consumption [kB] is given on the right hand axis. Six random datasets (sampled from a uniform distribution) of lengths ranging from 50 to 1000 are used.

this paper uses the scale function

$$k(q) = \frac{\delta}{2\pi} \sin^{-1}(2q - 1), \tag{6}$$

where $q$ is the quantile, $k$ is the scale function and $\delta$ is the compression parameter. Dunning and Ertl (2019) Figure 1 provides a visual representation of the scale function which shows a steeper gradient of $k$ near $q = 0$ and $q = 1$, reducing the cluster weights and increasing the accuracy at the tails. This compression parameter does not affect the shape of the scale function but it does increase the range of $k$. The greater the range of $k$ the more clusters will be used to represent the dataset, meaning a small value of $\delta$ will lead to less clusters, less accuracy and more memory saving and vice-versa for a higher $\delta$. These clusters can then be converted either to histograms (where bin densities may have non-integer values due the underlying cluster representation) or percentile estimates of the distributions.

The effect of the compression parameter $\delta$ on the memory requirements is shown in Fig. 3 (the effects on accuracy of the percentile estimates are given in Sects. 5.2 and 5.3). Here, six random datasets (from a uniform distribution) of different lengths are used to show how many clusters are required to represent the data as a function of $\delta$ (within the range $20 \leq \delta \leq 140$). Beyond a sample size of approximately 350, shown by the darkest three samples, the number of clusters used to represent the data grows linearly with $\delta$. This means that, for the range of $\delta$ tested, for all datasets with more than 350 values, the number of clusters is independent of the size of the dataset and is set only by $\delta$ i.e., a dataset of 500 values will be represented with the same number of clusters as a dataset of 5 million values.

For smaller dataset sizes (anything below 350, as shown by the three lighter blue / green samples on Fig. 3) the limit on the number of clusters may be set by the number of samples in the dataset. For example, beyond $\delta = 40$, the maximum number of clusters used to represent the the shortest (light green) dataset is 50. The number of clusters cannot grow anymore as it is already the length of the dataset meaning the distribution is represented in its entirety, with each cluster containing one





data sample and a weight of one. As the length of the dataset grows, the number of clusters also increases until it reaches the maximum limit defined by $\delta$ and shown by the larger datasets in Fig. 3. The key message to take away from this analysis is that,

for datasets containing less than 350 samples, there may be no memory savings generated from using this algorithm, as each cluster requires two values (mean and weight) for its representation. For example, when considering $\delta = 140$ and a sample size of 350, 180 clusters are used, requiring 360 values, which exceeds the length of the original dataset. However, for longer sample sizes and/or smaller $\delta$, the memory savings generated are substantial.

In the following Sects. 5.2 and 5.3, we examine the use of the t-digest algorithm with two case studies; wind energy in

Sect. 5.2 and extreme precipitation events in Sect. 5.3. These two examples have been chosen to examine how well the t-digest algorithm represents both the middle of a distribution around the median percentiles and its ability to capture extreme events at the tails of a heavily skewed distribution.

## 5.2 Wind energy

We present an application of the t-digest in the context of wind energy. With the decarbonisation of the energy system turning

into a global necessity, renewable energies such as solar and wind are becoming major contributors to the power network (Jansen et al., 2020). According to the Agency (2024), global installed capacity of wind energy, including both offshore and onshore resources, is expected to reach 1.72 TW by the end of 2028. However, unlike with fossil fuels, wind energy production is heavily affected by atmospheric conditions, subject to both short-term variability (i.e., weather), and longer term variations caused by seasonal and/or interannual variability (Grams et al., 2017; Staffell and Pfenninger, 2018). This volatility makes the

integration of wind energy into the power network a challenging task (Jurasz et al., 2022).

Having access to histograms of wind speed at high frequency (i.e., at hourly or sub-hourly scale) and hub height (i.e., height of the turbine rotor) is among the requirements of wind farm operators and stakeholders to estimate the available wind resources at a particular location. This information can be combined with the power curve of the turbines installed at each location to give reasonable estimates of energy production over a period of time. Obtaining an accurate representation of the wind distribution

is therefore crucial, as they condense the information from climate simulations required by the wind energy industry and aid in both the understanding of future output from current farms and in the decision-making relating to the viability of a proposed wind farm location (Lledo, 2019). Currently, there are two main methods for describing wind speed distributions; through full histograms of time series data or through fitting probability distribution functions to the data (Morgan et al., 2011; Shi et al., 2021). While the non-parametric approach (time series) generally outperforms the parametric (statistical) one in accurately

characterizing the distribution (Wang et al., 2016), it poses numerous challenges attributable to the large amounts of data required (Shi et al., 2021).

Here we investigate the use of the t-digest algorithm to estimate the wind speed distribution from streamed climate data. We use again data from ECMWF's IFS model (experiment tco2559-ng5-cycle3), this time looking at the 10 m wind speeds over December 2020. We again use the hourly data at native spatial resolution ($\sim 0.04°$), resulting in a global map containing

approximately 26.31 million spatial grid cells, 744 time steps and a full size of 145.82 GB with double precision (float64). Wind speed is calculated from the root of the sum of the squares of the 10 m hourly-mean northward and eastward horizontal




components. We conduct a detailed analysis on two locations, the offshore Moray East wind farm, located at (58.25 °N, 2.75 °E) in the North Sea, and the onshore Roscoe wind farm, positioned at (32.35 °N, 100.45 °W) in Texas. Both are marked on the global map in Fig. 4(a) in red and pink respectively.

Figure 4 shows a detailed comparison between how the t-digest and the conventional method (implemented using the numpy package in Python) describe a distribution. For the t-digest method, we used $w = 1$, meaning each $x_n$ was added to its' respective digest consecutively, to simulate data streaming. We start by examining the quantile-quantile plots in Fig. 4(b) and (e). Here the numpy percentile estimate is compared to the t-digest estimate (using $\delta = 60$) for all percentiles ranging from 1 to 100. The grey shaded areas indicate the range of wind speeds in which most commonly used turbine classes operate (Lledo,
2019). For the off-shore location in the North Sea, the extreme maximum percentiles lie outside the grey shaded region, but the lower tail is within it, whereas for the on-shore location in Texas the opposite is true for the shaded region. This shows that an accurate representation of the full wind speed distribution is required in order to cover the typical range of wind speeds relevant to wind farms. Both (b) and (e) show an almost perfect linear relationship. The main difference between the two lies in the storage requirements, with 5.95 kB needed to store the full 744 value time series of one grid cell used to generate the
numpy distribution estimate, compared to the 1.28 kB storage for the t-digest estimate based on 80 clusters ($\delta = 60$).

    Given such a high level of accuracy achieved with $\delta = 60$, we further investigate the effects of compression in Figures 4(c) and (d). For the same two locations, the difference in the estimate of the 50[th] and 80[th] percentiles between the two methods are shown as a function of $\delta$. The difference is represented as a percentage of the numpy percentile estimate calculated using the default linear interpolation method. The error bars represent the range of possible differences between the t-digest and
the numpy estimates obtained from using eight of the available interpolation schemes. These different interpolation schemes, outlined in Hyndman and Fan (1996), will provide a larger range in a percentile estimate if the data points are more sparsely distributed in the region of interest. Therefore, when any type of interpolation is required to estimate a percentile, there will always be a range a possible values depending on the interpolation method chosen. These error bars show how the t-digest compares against different methods available in Python. We see that the difference for the Roscoe wind farm in Texas, while
incredibly small, is slightly higher for both the 50[th] and 80[th] percentiles. This is due to the shape of the two distributions, as evident in the histograms in (b) and (e). Although the Moray East dataset has a larger variance, it more closely resembles a normal distribution, whereas the dataset for Roscoe is more uniform with the peak skewed to the left. The shape of the scale function given in eq. (6) will result in clusters with the lowest weight representing the distribution tails, while the clusters representing the middle of the distribution will have a larger weight. This is a clever characteristic of the algorithm, as due to
the bulk of the data in a normal distribution being centered around the median, these middle clusters can afford to be larger and cover a broader range of values without impacting precision. As the data from the Roscoe site slightly deviates from this normal distribution, a small increase in the difference is observed.

    However, despite this perceived higher difference for the Roscoe wind farm in Texas, the maximum differences for both the 50[th] and 80[th] percentiles are approximately 2% and 0.6% respectively, decreasing significantly as we increase $\delta$. Converting
these differences back into absolute terms, the maximum differences (at the lowest values of $\delta$) are 0.075 and 0.1 m s$^{-1}$ respectively, errors which would be considered extremely small for the end users. Moreover, while the 50[th] percentile estimate




**Figure 4.** (a) Global map showing the absolute mean difference between the t-digest ($\delta = 60$) and numpy (using the linear interpolation method) estimate of all wind speed percentiles from 1 to 100 given as a percentage of the numpy value. Wind speed hourly data across December 2020 from the IFS model. The map highlights two wind farm locations, one off-shore called Moray East in the North Sea (58.25 °N, 2.75 °E), marked with a red dot and another on-shore called Roscoe in Texas (32.35 °N, 100.45 °W), marked with a pink dot. (b) The quantile-quantile plot for percentiles 1 to 100 calculated using Python's numpy and the t-digest algorithm (with $\delta = 60$) for the off-shore location in the North Sea. The black dashed line represents the one-to-one fit, while the grey shaded area shows the range of wind speeds that most commonly used turbines operate in (Lledo, 2019). The upper light blue histogram is made using the t-digest algorithm ($\delta = 60$) while the darker blue is made using the built-in Python function. (c) The difference between the t-digest and numpy calculation of the 50th percentile, given as a percentage of the numpy value, as a function of $\delta$ for both marked locations. The error bars show the possible differences when employing all available numpy interpolation schemes, rather than the default linear interpolation method. (d) The same information as (c) but showing the difference for the 80th percentile. (e) The same as (b) but for the on-shore location in Texas.



at the Roscoe wind farm has the largest differences across all compression factors (light pink data in (c)) it also has the largest error bars, indicating that the conventional method also contains greater uncertainty. This highlights that the different interpolation methods used by numpy have a larger impact on the given result due to the sparser data, also explaining why the discrepancy between the one-pass and conventional methods is higher for this percentile estimate.

To ensure that the low differences shown in 4(b)-(e) are not specific to the two chosen locations, we calculated the absolute mean percentile difference (average across all percentile estimates from 1 to 100 i.e., over the quantile-quantile plots) for every global grid cell for $\delta = 60$, the results of which are shown in 4(a). Again, the data has been regridded to 1° for plotting convenience. In this global map, no difference exceeds 0.9% of the numpy value. Converting to absolute terms, this translates to no mean difference exceeding 0.068 m s$^{-1}$ across the globe. Taking the global spatial average of these mean differences gives 0.020 m s$^{-1}$. Based on this analysis, and the observed asymptote in the difference around $\delta = 60$ in 4(c) and (d), we note that further increasing the compression parameter would not significantly enhance the accuracy of the results for the wind speed distribution. Indeed, given the extremely small calculated difference, using a $\delta = 40$ would likely be sufficient to capture the distribution of global wind speed data required for users. Overall, as wind speed is best described by a Bi-modal Weibull distribution (Morgan et al., 2011), for monthly wind speed data at hourly time steps, the t-digest with $\delta = 60$ is more than suitable to fully represent the overall distribution while reducing the overall size of the global monthly data from 145.82 GB to ~33.2 GB. Looking at the monthly time series in one grid cell, this is compressed from 5.9 kB to 1.2 kB. For $\delta = 40$, this would reduce further to 0.85 kB.

This analysis has been conducted on monthly wind speed data, consisting of 744 samples (i.e. hourly values). With the compression of $\delta = 60$, we obtain an approximate five-fold reduction in memory requirements. If we were interested in a longer time series, for example annually, the hourly time series for one grid cell would require 8760 values ($\sim 70$ kB), while its representation with the t-digest would still only require 1.2 kB. On the contrary, if our interest were in weekly datasets with a time step of 1 hour, containing only 168 values ($\sim 1.3$ kB), using $\delta = 60$ would would still require 1.2 kB. Here no significant memory savings would be obtained, although the histograms could still be provided in real time to the users.

## 5.3 Precipitation

In the following Sect. we focus on the t-digest algorithm in the context of extreme precipitation events. It is necessary to examine these extreme events as intense rainfall and potential flood risk pose great social, economical and environmental threats. Both theory and evidence are showing that anthropogenic climate change is increasing the risk of such extreme events, especially in areas with high moisture availability and during tropical monsoon seasons (Gimeno et al., 2022; Thober et al., 2018; Donat et al., 2016; Asadieh and Krakauer, 2015). The need for climate adaptation measures in vulnerable communities exposed to these risks is pressing and, as with the other use cases in this paper, an accurate representation of the hazard is essential. Consequently, our focus here is on assessing how accurately the t-digest algorithm captures extreme events associated with the upper tail of precipitation distributions.

In this analysis, we use data from the ICOsahedral Non-hydrostatic (ICON) model (Jungclaus et al., 2022; Hohenegger et al., 2023) (experiment ngc2009) looking at precipitation over August 2021. We use half-hourly data using the Healpix spatial grid





(Gorski et al., 2005) ($\sim 0.04°$) containing 20.9 million grid cells. The full monthly dataset for this variable, containing 1488 time steps, requires 116.25 GB of memory using single precision (float32). The precipitation flux, given in kg m$^{-2}$ s$^{-1}$, has been converted to mm day$^{-1}$ for greater clarity. In the same way as the other sections, for the one-pass method we process the data sequentially with $w = 1$ to mimic data streaming.

Figure 5 illustrates the comparison between numpy and the t-digest in their estimates of the 99$^{th}$ percentile, with a focus on four specific locations characterized by different precipitation distributions. The chosen four locations, shown by different shades of pink in Fig. 5(a), represent a range of different precipitation distributions over this period and the locations in Brazil and the North Pacific have been specifically chosen to highlight the areas of largest discrepancy between the one-pass and conventional methods. A common theme amongst all of these distributions is that they are heavily skewed, with the majority

of the data falling around zero when there is very little to no precipitation. The dark red histogram in 5(c) represents a location in Brazil (13.50 °S, 60.00 °W) and shows an extremely dry month with almost all of the values in the 0 and 1 mm day$^{-1}$ bin (notice the logarithmic scale). We note there is a non-integer number of samples in some of the histogram bins. These non-integer values are due to a weighted contribution from the clusters to the histogram. The histogram in 5(d) is for a location in Colombia (4.50 °N, 78.00 °W) and depicts heavy precipitation over the month with maximum values above 400 mm day$^{-1}$ and

a more even spread across the distribution. The distribution over the location in the North Pacific (25.50 °N, 142.00 °E), shown in 5(e), also represents a large range of precipitation over the month with high maximum values, however the distribution is significantly more skewed to the first bin compared with 5(d). The final location, for Pennsylvania in North America (40.50 °N, 75.00 °W), shown in 5(f), has a lower range of precipitation values and again has the vast majority of the samples located in the first bin.

Figure 5(b) shows the absolute difference between the numpy and t-digest estimate of the 99$^{th}$ percentile for the total precipitation as a function of $\delta$. The corresponding number of clusters for each $\delta$ is indicated in grey along the upper axis. These differences are given on a log scale, as the North Pacific location has larger absolute differences compared to the other locations. This location was chosen explicitly to show where the one-pass and conventional estimates supposedly deviate. In this case (as with many locations with high absolute differences that were examined), due to the majority of data sitting around

zero, there are only a few values that comprise the remainder of the distribution. Due to this sparseness, the 99$^{th}$ percentile estimate falls in-between two data points, so the interpolation method used significantly impacts the estimate. This is reflected in the error bars for the North Pacific location in 5(b). While the absolute difference between the t-digest and the numpy estimate using linear interpolation is large, other numpy interpolation schemes result in a negative difference (not shown due to the logarithmic scale), showing that the t-digest estimate lies in-between the different estimates obtained with the available

numpy interpolation schemes. These larger differences can therefore be better attributed to the low density of values at the upper tails as opposed to poor representation of these tails by the t-digest. The other three locations show absolute differences around 1 mm day$^{-1}$ (which is considered negligible for most applications) with a decrease in difference as $\delta$ increases. These locations also show large error bars, again due to the highly skewed distribution and impact of interpolation, as was seen in the North Pacific location.





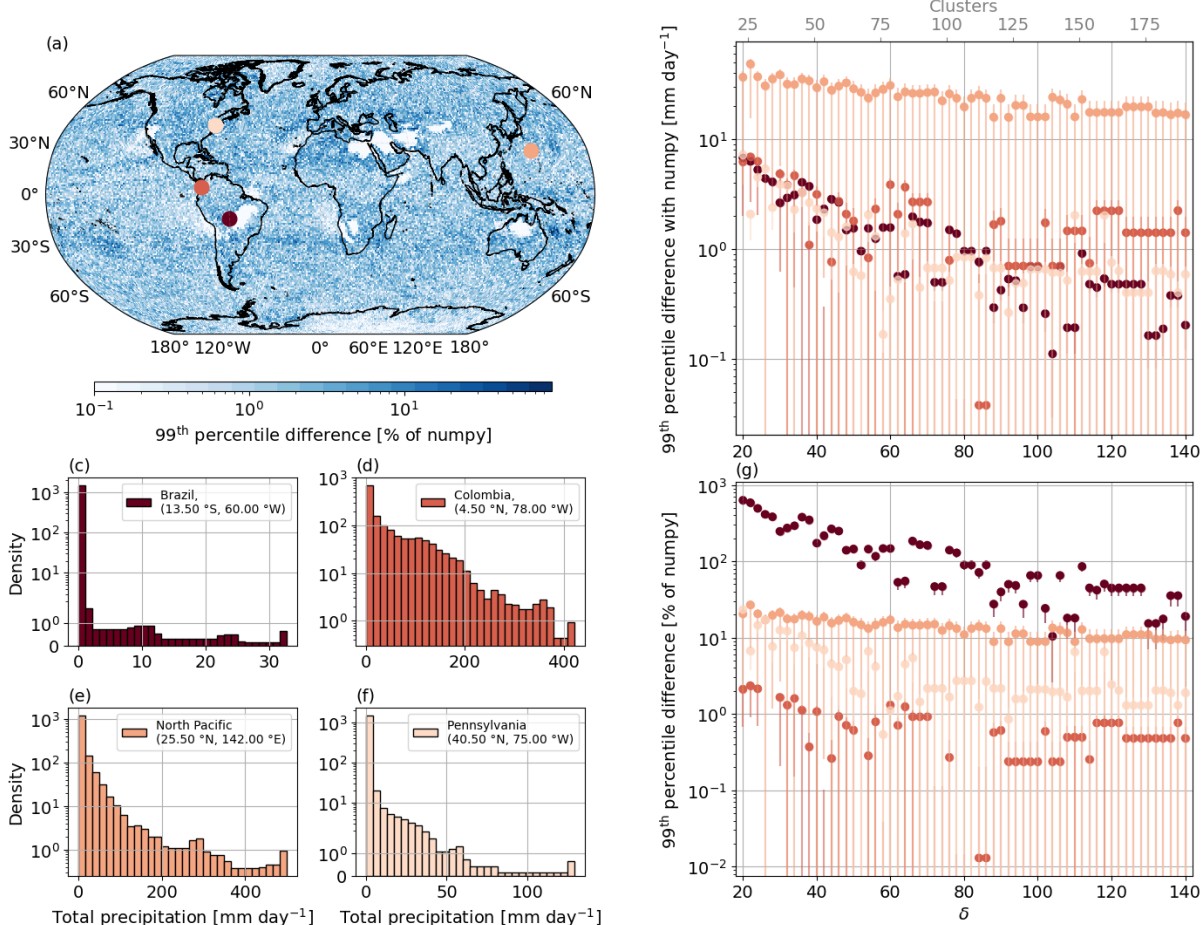

**Figure 5.** (a) The global map showing the absolute difference between the t-digest ($\delta = 80$) and numpy (using the linear interpolation method) estimate of the 99$^{\text{th}}$ precipitation percentile given as a percentage of the numpy value. Precipitation half-hourly data from the ICON model (experiment ngc2009) over August 2021. The locations of the four solid pink circles are given in their respective histograms. (b) The absolute difference between the numpy and t-digest 99$^{\text{th}}$ percentile estimate shown as a function of compression for the four marked locations on (a). The upper grey axis shows the number of clusters used in each digest for each $\delta$. The error bars represent the range of possible absolute differences based on all available numpy interpolation schemes, in contrast to the default linear interpolation method. (c)-(f) Histograms showing the distribution of the total precipitation (mm day$^{-1}$) at all four location for their respective August 2021 time series. Each histogram color corresponds to its location on the global map and the histograms are calculated using the t-digest with $\delta = 80$. (g) Same as (b) but given as a percentage of the numpy value.

Figure 5(g) shows the same differences presented in 5(b) but given as a percentage of the numpy estimate. Again this is shown on a logarithmic scale due to an increase (in some cases by orders of magnitude) in the difference for some locations when given as a percentage, as shown by the Brazilian location in dark red. This dramatic increase is due to the 99$^{\text{th}}$ percentile estimate





being so close to zero for distributions with almost no precipitation, such as the one in Brazil. In the t-digest representation of these distributions, all of the streamed data points with values very close to zero are placed in a few clusters at the tail of
the distribution where they then are averaged. As the granularity inside the cluster has then been lost, the percentile estimate will be inaccurate. In absolute value terms this is in inconsequential as the values are so close to zero. Indeed for the Brazilian location the absolute estimate of the $99^{th}$ percentile is extremely close to that of numpy and lies around 1 mm day$^{-1}$ When represented as a percentage however, the difference can be greater than 100. To account for these unrealistically poor estimates we calculated the percentage difference for both 5(g) and 5(a) using error $= 100|(a-b)|/(b+\epsilon)$ where $a$ is the t-digest estimate,
$b$ is the numpy estimate using the linear interpolation (other than the error bars in 5(g)) and $\epsilon = 1$. This small constant $\epsilon$ is introduced to stabilise the calculation when $b$ is extremely small. The results are shown globally (using $\delta = 80$) in 5(a), again on a logarithmic scale. The data has been regridded to $1°$ for plotting convenience. Most of the differences fall between $1\%$ and $10\%$ of the numpy value, however due to the reasons just described some differences are unrealistically large. The average of the global differences are shown in Table 1 as a function of $\delta$, given as both the percentile estimate and the absolute value.
We have included this table to highlight that, due to the extremely low precipitation values for some of the estimates, a higher percentage difference does not necessarily translate to a higher difference in absolute terms. Even with a relatively small $\delta$ of 40 the overall relative difference of the $99^{th}$ percentile (when averaged globally) is less than $4\%$ (absolute difference of 1.27 mm day$^{-1}$). Both of these differences decrease to less than $2\%$ and 0.60 mm day$^{-1}$ as $\delta$ increases to 120.

**Table 1.** Global average of the $99^{th}$ percentile difference as a function of compression, given as both the absolute difference in (mm day$^{-1}$) and as a $\%$.

| $\delta$ | $\|(a-b)\|$ (mm day$^{-1}$) | $100\|(a-b)\|/(b+\epsilon)$ (%) |
|---|---|---|
| 40 | 1.27 | 3.77 |
| 60 | 0.91 | 2.63 |
| 80 | 0.75 | 2.14 |
| 100 | 0.65 | 1.86 |
| 120 | 0.60 | 1.67 |

Based on the results from Table 1 and Figs. 5(b) and (g) we see a reduction in difference with increasing compression that
asymptotes around $\delta \approx 80$. This is higher than the asymptotic value seen in Figs. 4(c) and (d) of approximately $\delta \approx 60$. In general, there are no significant accuracy improvements from using a $\delta$ more than approximately 80 (100 clusters) to represent the precipitation distributions and we would not recommend exceeding this compression factor. Indeed, depending on the specific accuracy requirements it may be beneficial to reduce this even further. Based on a the results from $\delta = 80$, the entire dataset of 116.25 GB could be represented with 36.57 GB, reducing the memory requirements by approximately a third.
One interesting point to raise is how the scale function for the t-digest algorithm - given in eq. (6) - impacts these results. While the differences obtained for the precipitation distributions are well within the acceptable limits for most use cases, comparing them with the results in 5.2, we see poorer accuracy. This is due to the wind distributions more closely resembling





a normal distribution, which is the distribution that the symmetric scale function describes best. While outside the scope of this investigation, we note that to better represent these skewed precipitation distributions, a non-symmetric scale function

that would create larger clusters at the lower tail may more accurately capture the underlying distribution. Another method to improve the representation of the dataset would be to simply impose a cut-off (such as 1 mm day$^{-1}$) for the data that is added to the digests. Removing this extremely larger cluster close to zero would, in many cases, improve the representation of the data by the t-digest.

## 6 Convergence

So far we have considered the comparison between the output from the one-pass algorithms vs their conventional equivalents at the end of the full dataset ($n = c$), i.e. how well do they represent the final statistic. However there is an additional aspect of one-pass algorithms that requires consideration: the concept of convergence. These one-pass algorithms provide a rolling summary $S_n$ of the statistic after every time chunk, offering potential value to certain applications. For example, when performing bias-adjustment on climate models, it is necessary to compare the model climatology (or probably distribution function) against a

reference climatology of a certain area. Bias adjustment is often based on these probability distribution functions. In streaming mode, we use the t-digest method to build this model probability distribution function, which will evolve as we add more samples. Knowing how many samples need to be added to the t-digest (or rolling summary $S_n$) until it accurately represents the distribution is highly valuable to these bias-adjustment algorithms within the context of streamed climate data. The bias adjustment is only effective after enough samples have been added and adjusted, before this, $S_n$ should not be used for further

analysis.

This concept of convergence is explored by examining the rolling summary $S_n$ for 2 m temperature, 10 m wind speed and precipitation flux from the IFS and ICON models. The temperature and wind speed datasets are the same IFS datasets used in Sects. 3 and 5.2 respectively and the precipitation dataset is the same ICON data used in Sect. 5.3. For both temperature and wind speed the rolling summary $S_n = \bar{X}_n$, while for precipitation $S_n$ is the rolling 50$^{\text{th}}$ percentile estimate. For all rolling

summaries $w = 1$, meaning that the number of samples, $n$, contributing to $S_n$ grows by one each time. Unlike in the previous Sects. where we were interested in the value of $S_n$ at $n = c$, here $S_n$ is stored at every time step, providing a time series of its development denoted as $\mathbf{S}_n = \{S_1, S_2, ..., S_n\}$. The rolling standard deviation ($\sigma$) is then taken of $\mathbf{S}_n$ using eq. (4) and eq. (5). This results in a time series of $\sigma$ defined as $\boldsymbol{\sigma}_n = \{\sigma_1, \sigma_2, ..., \sigma_{n-1}, \sigma_n\}$, where, for example, $\sigma_{n-1}$ represents the standard deviation of the series $\mathbf{S}_{n-1}$.

The outer axis in Fig. 6 shows $\boldsymbol{\sigma}_n$ over time using the temperature data at four locations marked in the legend. As expected, Fig. 6 shows how the series $\boldsymbol{\sigma}_n$ for the different datasets are represented - after an initial peak - by inverse exponential asymptotes when plotted over time. $\sigma_n$ decreases as the distribution represented by $S_n$ stabilizes due to the addition of more samples. The data shown here uses temperature data but results using precipitation or wind speed data follow the same shape, just with different maximum peak values.



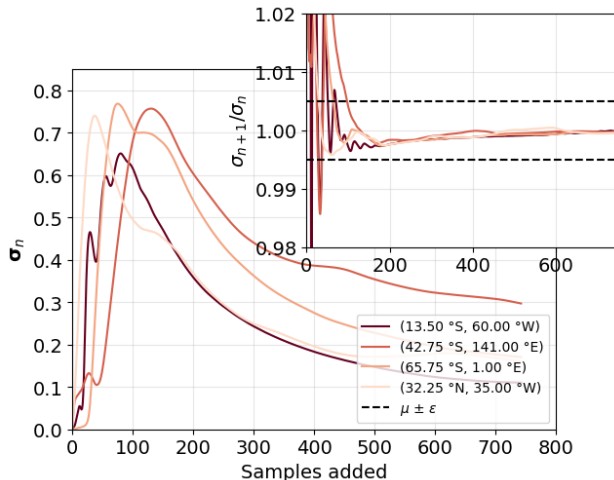

**Figure 6.** The convergence of the series $\boldsymbol{\sigma}_n$ for the temperature dataset. The outer axis shows four $\boldsymbol{\sigma}_n$ series where the location of the points are given in the legend. The inner grid shows the result of the left hand side of eq. (9).

The inner axis of Fig. 6 then shows the time series of $\sigma_{n+1}/\sigma_n$ of the four datasets. We plot this standard deviation ratio to define when the time series have converged, i.e., when sufficient samples have been added to $S_n$ that the standard deviation of the series $\mathbf{S}_n$ has stabilised. This is formally defined using the order of convergence definition

$$\lim_{n \to \infty} \frac{|\sigma_{n+1} - L|}{|\sigma_n - L|^q} = \mu, \tag{7}$$

where $q$ is the order of convergence, $L$ is the convergence limit, $\sigma_n$ is the standard deviation of the $\mathbf{S}_n$ time series at time $t = n$

and $\mu$ is the rate of convergence (Grau-Sánchez et al., 2010). As both $q$ and $\mu$ are unknown in eq. (7) the order of convergence was estimated using

$$q \approx \frac{\log \left| \frac{\sigma_{n+1} - \sigma_n}{\sigma_n - \sigma_{n-1}} \right|}{\log \left| \frac{\sigma_n - \sigma_{n-1}}{\sigma_{n-1} - \sigma_{n-2}} \right|}. \tag{8}$$

Interestingly, when eq. (8) was calculated over $\boldsymbol{\sigma}_n$ for a random selection of grid points the approximation of $q$ (for all variables) was centered around 1. This indicated a linear convergence rate for all the standard deviation time series over the global grid

cells. Using this approximation and taking $q = 1$, $L = 0$, eq. (7) could be reduced to

$$\frac{|\sigma_{n+1}|}{|\sigma_n|} = \mu + \epsilon, \tag{9}$$

where we have added the small parameter $\epsilon = 0.005$, which defines the boundaries of convergence, as shown by the black dashed lines on the inner axis of Fig. 6. Once the $\boldsymbol{\sigma}_n$ series fell within this range (and did not leave again) then $\boldsymbol{\sigma}_n$ was considered to have converged. We emphasise that this analysis is not saying the standard deviation of the series $\mathbf{S}_n$ is no

longer changing. These climate variables will exhibit long time-scale temporal variability due to climate variability and change and we accept that their mean values and 50th percentile estimates will shift over time. What these results show is how many




samples are required to contribute to the one-pass summary $S_n$ until we can consider it an accurate representation of the overall distribution at that point in time.

This analysis is shown globally in Fig. 7. In Fig. 7(a), (c) and (e) we show the global standard deviation $\sigma_n$ of $\mathbf{S}_n$ at $t = c$ for temperature, wind and precipitation respectively. Here $c = 744$ for the hourly temperature and wind speed data, while $c = 1488$ for the half-hourly precipitation data. As expected, the standard deviations for all the variables are larger in areas that experience more variability. For example, the final standard deviation of the rolling temperature mean shown in (a) shows much larger values away from the equator where temperature averages will experience seasonal variation. Figure 7(b), (d) and (f) then show the the number of samples required for the standard deviation $\sigma_n$ of these rolling summaries to converge, as defined in (9) when $|\sigma_{n+1}|/|\sigma_n|$ falls within the range $\mu \pm \epsilon$.

Unexpectedly, the convergence time does not have a strong correlation with the final standard deviation $\sigma_n$ shown on the left column. This is re-assuring, as it shows that convergence is partially independent of the actual spread of the data and allows conservative estimates of sample size to be used for global data. What is particularly noteworthy is that, across all these datasets, the number of samples required for convergence is extremely similar. Taking the mean value of Figs. 7(b), (d) and (f) results in 82, 77 and 81 samples respectively. This is striking, especially given that Fig. 7(f) represents the convergence of the 50[th] percentile estimate, as opposed to the mean in Figs. 7(b) and (d). This indicates that, for hourly and half-hourly data, approximately 4 days are required to accurately represent the month.

It should be mentioned however that this method is used as a general test of convergence for specific datasets. This analysis was also carried on out on observational data sets (not presented), using both monthly and daily time steps, as opposed to the climate model hourly and half-hourly data shown here. For these data sets the average number of samples required for convergence was approximately 50 when averaged globally. As the values of this observational data were already given as daily or monthly summaries, extreme events (such as wind gusts) were already smoothed in the original time series. The better the representation of shorter extreme events in the original data, the longer we would except for the data to 'converge', as these values will not have been pre-averaged. This is highlighted in 7(f), which uses the half hourly precipitation data. Even though the average number of samples required is 81, there are some areas where 600 samples are needed before convergence, approximately double the maximum shown in 7(b) and (d). Due to the extremely short time step of this data, extreme events will be better represented. Overall, in this Sect. we have presented how the one-pass algorithms provide added value in the context of bias-adjustment for streamed climate data. We present a criteria for stabilisation that we use to define how many samples are required to be added to a rolling one-pass summary $S_n$ before that summary can be used as a representation of the whole distribution.

# 7 Conclusions

Within the climate modelling community the generation of increasingly larger data sets from higher resolution GCMs is becoming almost inevitable. While there is clear argument for the added value of these high resolution models, new challenges of handling, storing and accessing the resultant data are emerging. One novel method being investigated by the DestinE project





**Figure 7.** (a) The standard deviation of the full $\mathbf{S}_n$ time series at the end of the IFS monthly temperature time-series during March 2020. Here $S_n = \bar{X}_n$. (b) The number of samples required (i.e. number of time steps) that it takes for the rolling standard deviation of the $\mathbf{S}_n$ time series to converge. Convergence is defined in the text. (c) Same as (a) but for the IFS monthly wind speed time series over December 2020. (d) Same as (b) but for wind speed. (e) Same as (a) and (c) but here $S_n$ is the estimate of $50^{\text{th}}$ using the ICON precipitation time series over August 2021. (f) Same as (b) and (d) but for precipitation.



(Bauer et al., 2021; Hoffmann et al., 2023) is data streaming, where climate variables at native resolution are passed directly to climate impact models in near model run time. This article has presented algorithms designed to handle this streamed climate data. The application of each algorithm has been demonstrated through relevant use cases in the context of climate change impact studies. We categorized the statistics into two different classes, ones that can be represented by a single floating-point value, and those which require a distribution. For those that require only a single value (e.g., mean, standard deviation, mini-

mum, maximum, threshold exceedance), we obtain accuracy at the order of the machine precision, well beyond the accuracy required or indeed provided by climate models themselves. While providing the same result as the conventional method, these algorithms allow the user to keep only a few rolling summaries in memory at any one time. Unlike a conventional statistic, where the memory requirements for computation scale linearly as the time series grows, the one-pass algorithms for these statistics provide an easily implemented, user-oriented method that bypasses these potentially unfeasible memory requirements.

For the statistics that require a representation of the distribution (e.g. percentiles and histograms), we applied the t-digest method, framed within relevant use cases. In Sect. 5.2 we focused on wind, a variable which requires an accurate representation of the full distribution in the context of renewable energy. Using a compression factor of $\delta = 60$ (approximately 80 clusters), the mean absolute percentile differences for global monthly wind speeds did not exceed 0.9% of the estimate given by the conventional method. For precipitation, given in Sect. 5.3, we focused on the extremes of this skewed precipitation distribution.

Due to the presence of low frequency extreme events there was more discrepancy between the numpy and t-digest estimates. In the cases of high absolute difference between the two estimates, there were also large error bars from the different interpolation schemes of numpy. Examining these distributions showed these higher differences were due to sparseness of data in the distribution as opposed to poor representation from the t-digest. In the case of high percentile difference, these were unrealistic differences that occurred due to division by extremely small numbers generated from the numpy estimate and also occurred

when precipitation fell around 0 to 1 mm day$^{-1}$, negligible values in terms of the user interested in extreme rainfall events. Overall, when averaging the differences globally, we obtained (for $\delta = 60$) 2.63% or 0.91 mm day$^{-1}$ in absolute terms.

    In both the wind speed and precipitation analyses, increasing the compression factor and using more clusters to represent the distribution did not necessarily result in higher accuracy. Due to the larger memory requirements at higher compressions and with achieving such accurate representations for both wind speed and precipitation, distributions with $\delta = 60$ and $\delta = 80$

respectively, we would not recommend using larger compression factors.

    Overall, we have demonstrated the effectiveness of one-pass algorithms on streamed climate data and provide their Python implementation ready for use in data streaming work flows (Grayson, 2025). These algorithms not only provide accuracy well within the required limits of climate model variables but also empower users to harness the full potential of high-resolution data, both in space and time. Indeed, while some of the methods contain small errors (specifically the t-digest) we note that not

harnessing the added value of high-resolution data due to storage limitations will also incur a potentially more significant error. Due to the fact that only a few rolling summaries are required to be kept in memory, these statistics become time independent, allowing users dealing with high-resolution GCMs to select any variables at their native resolution and process them in next to near model runtime. This eliminates the constraints of relying on pre-defined archives of set climate variables, typically provided months to years after the models have been run. With the continuing movement to higher resolution, the streaming of

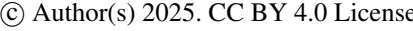


climate data will become a fundamental paradigm of data processing. This paper showcases the features of one-pass algorithms across a range of relevant statistics that can be harnessed to work in the new era of data streaming.

*Data availability.* Full data from the nextGEMS cycle 3 are openly accessible and can be found at NextGEMS (b), including output from the development Cycle 3 (Koldunov et al., 2023) and production runs (Wieners et al., 2024) for both ICON and IFS. All the nextGEMS netCDF data used to demonstrate the use of the one-pass algorithms described in this paper (e.g. looping through the time dimension to simulate data 505   streaming) and to create all the figures in the study are available at the Zenodo repository 'nextGEMS cycle3 datasets: statistical summaries for streamed data from climate simulations v3' with https://doi.org/10.5281/zenodo.12533197 and with Creative Commons Attribution 4.0 International license (Grayson, 2024b).

*Code availability.* Version 0.4 of the repository that contains the implementation of the one-pass algorithms described in this paper, as well as Jupyter notebooks to reproduce all the figures, is preserved at https://doi.org/10.5281/zenodo.12533064 and developed on Github (Grayson, 510   2024a). The source code for the one-pass package v0.6.2 implementation, ready for integration into data streaming workflows, can be found at https://doi.org/10.5281/zenodo.14591828 and licensed under Apache License, version 2.0 (Grayson, 2025).

*Author contributions.* Llorenç Lledó conceptualized the ideas behind the study while Katherine Grayson and Stephan Thober developed the methodology. Katherine Grayson conducted the formal analysis and wrote the original draft. Stephan Thober and Francisco Doblas-Reyes both supervised. Aleksander Lacima-Nadolnik and Ehsan Sharifi both helped validate the study. All authors contributed to the review and 515   editing process.

*Competing interests.* The authors declare that they have no conflict of interest.

*Acknowledgements.* We would first like to acknowledge Bruno Kinoshita for his help and advice on the t-digest algorithm. We would also like to acknowledge Paolo Davini and Jost Von Hardenberg for developing tools that allowed for all the data retrieval. We would like to acknowledge the Generalitat de Catalunya (ClimCat, ARD209/22/000001) and European Commission Destination Earth Program. 520   Destination Earth is a European Union funded initiative and is implemented by ECMWF, ESA, and EUMETSAT. We would also like to acknowledge the European Commission Horizon 2020 Framework Program nextGEMS, under grant number GA 101003470.



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
