# Peer review of "Statistical summaries for streamed data from climate simulations: One-pass algorithms"

_EGUsphere, 2025_

## Referee Comment (RC2)

Review of "Statistical summaries for streamed data from climate simulations: One-pass algorithms (v0.6.2)" by Katherine Grayson, Stephan Thober, Aleksander Lacima-Nadolnik, Ehsan Sharifi, Lorenc Lledo, and Francisco Doblas-Reyes

For consideration in *Geoscientific Model Development*

Recommendation: Major revision

This article describes techniques for on-line statistical analysis of climate model output which can reduce the amount of output data without needing to degrade the spatial resolution of the output. In particular a sophisticated, novel t-digest algorithm is described for on-line generation of percentiles and histograms. The publication is timely given the increasingly large volumes of output being created by increasingly complex and high-resolution numerical models, especially for the emerging class of km-scale models many modeling centers are developing. In particular the quantification of memory savings is a powerful selling point. This paper should be published and GMD is the ideal journal for this purpose. However I was somewhat confused by the discussion of the t-digest method and would like to have some more clarification on a number of points.

I am unfamiliar with some of the terminology. Does the "scale function" $k(q)$ introduced in equation (6) define the "edge values" for each of the bins/clusters? If so, an illustration (showing the domains for each cluster as a function of delta) would be useful. Also, should the same scale function be used for each variable? While having a somewhat equal-spaced set of bins is possibly a good choice for wind speed, this would not be as useful for precipitation since (as is shown in Section 5.3) small values are far more common than large ones, and so the small bin will fill up very quickly.

Also, how the t-digest method goes from its clusters to creating percentiles and histograms is unclear. Could a more concrete demonstration of this be shown? Also, are the numpy percentiles and histograms created by a similar algorithm, or are these the exact percentiles and histograms computed from all of the data without approximation?

I was very glad to see the careful validation of the simpler one-pass algorithms (mean, standard deviation) and the striking demonstration that the errors are at rounding level. I had more trouble understanding Figures 4 and 5. It is not immediately clear that the red and orange dots represent the different locations in panels (b)--(e) without a legend; and the histograms inset in panels (b) and (e) are so small as to be almost impossible to interpret. It would be more useful to overplot these on a larger panel to more directly compare the two. I also don't quite understand the main result in panels (b) and (e): the shaded area is the typical wind turbine operating tolerance for these locations, and each dot is another percentile larger (first dot is 1st percentile, last is 99th)? Overall I think splitting out the panels and/or insets into different figures, providing better labels and annotations, and enlarging some plots would make it much easier to read and understand.

The discussion of convergence of climatologies in Section 6 is interesting and does get at the vital scientific question of how much data is needed to create a useful climatology. However, given that the one-pass algorithms create nearly-exact statistics (for means and standard deviations) or up to the cluster's sampling uncertainty (t-digest) *for the model run segment being considered*; constructing a climatology for bias correction or other purposes would be a downstream step that would presumably mean aggregating statistics over a large number of segments and/or individual simulations. Would not this analysis then say how long the simulation needs to be to create an accurate climatology, and not have much effect on the algorithms of this paper?

Minor comments:

- There is precedent for on-line calculations of statistics such as means, extrema, standard deviations and so on. See the "data reduction" techniques here, for instance:
https://github.com/NOAA-GFDL/FMS/blob/main/diag_manager/diag_yaml_format.md
https://github.com/NOAA-GFDL/FMS/blob/main/docs/diag_table.md

- Line 171: Should "extremely values" be "extremely small values"?

---

## Author Comment (AC4)

**Response to Reviewer 1 - 2025-28**

**I.  REVIEWER SUMMARY**

In this manuscript, the authors present an alternative way to process "on-the-fly" (online) model output data, with the motivation being reduced data/memory usage and potential ease for users. Ultimately, this is an interesting manuscript, albeit weird. I think it can be made worthy of publication upon a revision. In general, the authors seem to be factual, careful, and nuanced. I think most of the work is relatively high quality. There are some problems though (the first one potentially fatal for the manuscript).

**II.  MAJOR COMMENTS:**

1. My reading of the manuscript is that it is making use of an algorithm package (whose version is in the title no less), but upon examination of the code availability section, I don't think the authors actually used the package at all in their analysis. Instead, they "simulated" using the package. I find this quite odd. I wrote a comment about this at the end. I don't know if this is intentional or not, but this is pretty deceptive. I invite the authors to explain, and I will keep an open mind.

   The reviewer raises a valid point here. To explain, the source code of the One_Pass package was not originally introduced due to contractual obligations. As we still felt that the analysis of the algorithm's performance was worthwhile, the backbone of the package was extracted so as to produce the same results, just not using the easy user interface and functionality of the One_Pass. This however was not deemed appropriate and when we were granted permission to make the package open source, we did so but did not update the corresponding notebooks. We have now updated the notebooks accordingly, in order to present the same Figures but produced with the code of the One_Pass package. We do note however that in some of the notebooks relating to the t-digests we also directly use the underlying t-digest package for analysis. This is because the One_Pass package acts as a user interface, hiding the background computation and will simply return the final percentile or histogram (depending on the user request). We do not expose the inner workings of the t-digest to the user. In the paper analysis however, we also provide details on cluster sizes and weights,

information not required when examining the final percentile but interesting for analysis and obtainable via the t-digest package. There was never an intent to be deceptive here and we have now more thoroughly integrated the One_Pass into the manuscript and used it in the making of the Figures.

2. The authors didn't explain how this type of algorithm/package would be used with an actual climate model running. Maybe it would be used in a futuristic cloud-compute setup? (They do cite some cloud-computing works.) I think this manuscript would greatly benefit from a detailed example (ideally a workflow) of how this would be implemented in an end-to-end fashion.

We agree that the manuscript would benefit from a description of the overall setup. Originally this was not included due to limitations of not allowing the package to be open source, so we kept the scope of the paper to the more theoretical implementation. After clearing the contractual limitations that originally lead to the exclusion of the source code of the One_Pass package, we have included a section 'Design and implementation into a workflow' (Sect 2.). Here we describe the design choices of the package in the context of the project that it was built for. We explain the basic configuration and how it is currently used in an HPC workflow via a workflow manager.

3. I think this manuscript would benefit from a comparison to the "online diagnostics" route. In the "online diagnostics" route, climate scientists (and developers) write functionalities to output specific items of interests, without needing too much data. Climate models already do online mean, min, max, etc. in their output streams — nothing here is novel. That could easily be extended to all sorts of statistics. In fact, it could be extended in a composable fashion to even more complex and variable algorithms. For example, consider the following: "I want to calculate the globally averaged temperature at cloud-top but with a threshold of 280 K". An algorithm can be written to identify cloud top, then finding the temperature there, considering the threshold, and then doing a horizontal reduction in a composable way. This algorithm can then be run after each time step in the model, and storing in memory the variable to be output or writing it out immediately. I think the authors could improve this manuscript if they compare and contrast their one-pass streamed approach to that of models natively doing more online calculations themselves (potentially on idling CPUs, if the models run on GPUs) instead of an external module/package.

While it is true that models often provide mechanisms to compute variable statistics during the simulation runs, the key here is that a model no longer needs to be tailored to the specific needs of a data consumer downstream. The increasing resolution of climate models introduces much complexity when trying to tailor the model outputs to the requirements of the data consumer. While it is common for models to produce online calculations of monthly means of certain variables, getting different modelling groups to align on specific statistics for downstream users (e.g. 99th percentile of wind speed) would be challenging, to say the least. In the new section we describe the design choices behind the One_Pass package, and that it needs to be able to integrate with a variety of GCMs. We also outline the concept of the ClimateDT and that one of the ideas is that data consumers may enter or leave the simulation data stream at any time, according to their needs, and do not enforce their needs onto the design of the model. Instead, with the One_Pass they are provided with a mechanism to select their fields of interest, as well as the aggregation method and frequency for those fields. This will make the necessary operations at a later stage downstream and will be performed by an independent process, making the whole ecosystem more scaleable. We hope the inclusion of the new section has given this more context and also addresses the minor comment 7 for for L35–42.

Minor comments

1. L2: nitpick: "as such" is kinda weird here in that it made me think running at high-res was a consequence of the preceding sentence, but it actually is more of a antecedent of the following sentence.

   We've removed this along with the first sentence in an attempt to condense and consolidate the abstract.

2. L5: nitpick: probably misplaced comma (I'd move it to after HPC instead)

   Fixed.

3. L7: the phrase "data streaming" is introduced like a known quantity that is specific climate science, but in fact, it is neither known nor usually associated with climate science. Maybe jargon? If I were you, I would consider removing "data stream;" (the phrase with the semicolon) and let the sentence stand without the disruption

Data streaming is now properly defined in the abstract and in the introduction. It is a key concept that requires explaining as without projects adopting this method, there is no need for one-pass algorithms or the package. We have made sure to better define all jargon used in the manuscript.

4. L9: one-pass may necessitate explanation here (so I would rephrase)

We've better defined these terms in the abstract and introduction making sure not to introduce jargon without explanation.

5. L9: intelligent doesn't seem like the right word (maybe efficient?) — these techniques aren't learning things, or are there?

You're right, they're not learning algorithms, we've removed this adjective in the rephrasing. Also removed in the introduction.

6. L17: I think "well within the acceptable bounds" is an underestimated. Imho and understanding, these algorithms basically recover accuracy to within an insurmountable precision limit, so basically as good as one would get anyway. I would rephrase to finish with a stronger point

Rephrased to make this point.

7. L35–42: I think this may surprise an average climate scientist involved in model development and evaluation (such as myself). We always had the ability to access data before simulations are done and one could obviously write stuff in a continuous manner. I think what you want to highlight is that you're algorithmically calculating some interesting statistics that capture specific interests. That is not really novel, not unique, but it is interesting and worthy of publication. In other words, I don't see the "novel" method and "unprecedented" reduction of "meaningful" output. Can you please (carefully) elaborate?

We understand the reviewers concern here and agree the paradigm that the One_Pass package has been built around has not been fully explained. Indeed, someone involved in model development can write and access the climate model data before the whole simulation is finished. Here we are referring to a more flexible set up where data consumers who are not part of the model development can still 'tap into' the data stream. This means someone who would traditionally access data via an online archive could embed into the data stream and

start working with the data without having to wait for the full model run to be finished and published online. To address your concern we have modified the introduction significantly to convey data consumers as being those who don't necessarily work in the model development and traditionally don't have access to model data as it's being produced. We have also included the new section 2 (already detailed above) which puts the development of the One_Pass into the context of the project, with the eventual aim being anyone with an impact model or similar can embed into the data stream.

8. L70: the last sentence here gives the wrong impression of what you're trying to say. Your manuscript is supposed to showcase those algorithms, so "requesting" readers to go follow some other package documentation may not be the best way to politely say, "We don't discuss the code implementation of specific algorithmic details related to each statistic, and we instead focus on showing their utility to climate analysis" . . .

We have more thoroughly embedded the differences between the One_Pass package and the aim of the paper, which as you say, is looking at the algorithms utility. We have rephrased L70 and tried to make it clear that examples of the package can be found online while here we discuss accuracy.

9. L194: I'd elaborate on the different results part

We have expanded this statement to provide some more details however when sampling other packages we did not find substantial differences between the implementations. Our decision was to use this particular implementation was also based on how often the package was being maintained.

10. L219: I think you can say that plainly at the outset without taking the reader on an unnecessary voyage. I am not sure if the added info (analysis) is informative otherwise. Feel free to disagree.

We realize this paragraph was quite convoluted. We have re-worded to introduce the main point at the start of the paragraph and have tried to condense to make the message clearer.

11. L268: so, the NumPy-calculated one is the "reference" right? (Note, I would refer to NumPy like they refer to it in their papers/docs, uppercase N and P)

Yes, NumPy is the reference. We've made this explicit in the introduction and fixed the casing.

12. Section 5.3: If I were you, I would simply delete this section. Your concluding paragraph (around L385) basically shows that most of the preceding analysis/"results" should be taken with a giant grain of salt. I would simply avoid the distraction, maybe add something about how the analysis in 5.2 could done on precipitation with a meaningful cutoff and/or a different type of underlying assumption (distribution type)

We don't agree that removing this section is a good idea. The aim of the paper is to show how well the package will represent different statistics for different climate variables that are of interest to the climate community. Accurately capturing extreme precipitation is essential and we want to provide a detailed analysis of how the package performs on this front. We have however made some modifications to the section, reducing the text to make it more conscience and updating the Figure. Now we show comparisons between the histograms for the t-digest method and the NumPy method which aid in the explanation of the differences for the 99th percentile estimate.

13. L431: But this doesn't make sense to me. As a scientist, what I care about is the stability of the value, not an arbitrarily defined metric of convergence. In Figure 6, $\sigma_n$ goes from roughly 0.35 at 200 samples to 0.15 at 600 samples, but it is within those "convergence" lines.

We use the convergence rate https://en.wikipedia.org/wiki/Rate_of_convergence to define how long it takes for the underlying statistic summary to be representative of the statistic of whole data set. We are not claiming the statistic has reached a final or static value. Climate variables will exhibit temporal variability and long-term trends, and we do not assume that their statistical properties remain fixed over time. Rather, our focus is on when the variability in the estimate of $\sigma_n$ itself has sufficiently diminished — i.e., when its fluctuations due to sample size limitations become negligible compared to inherent data variability. This is to allow the statistic summaries to be used in methods such as quantile-quantile mapping for bias-adjustment of streamed data. We have modified the introduction of the section considerably to clarify this.

14. L432: Ok, but how does that impact your assertions earlier about all the savings and not

having to wait for model simulations to run a lot of steps? I guess you're saying, one kinda has to wait a lot of steps to get "accurate" stuff out of these one-pass algorithms? Like 200 steps or so?

The analysis conducted here doesn't impact earlier assertions about the use of one-pass methods. As with previous sections, here we are benefiting from the memory reduction of the one-pass and the ability to work with the data stream. Our point about timely access was meant to compare, for example, those with impact models who currently use data from online archives. In the overall aim of the ClimateDT, these types of models will be able to embed into the ClimateDT workflow and access directly the model output via this streaming paradigm. We hope that with the inclusion of the new section this has been made much clearer. If you don't agree, we would greatly appreciate more thoughts on this.

15. Section 6: Like Section 5.3, I think the convergence analysis is likely misleading, potentially counterproductive, and perhaps better left out. Your goal in this manuscript is to showcase something useful for users of this method. My assumption is that if people choose to use this type of one-pass algorithm, they would do so on relatively high-frequency data and they would understand the risk of under-sampling. Maybe I am just not getting what you're trying to do here? Could you motivate it better? Is it anything other than something like "don't take the mean of the first 5 time steps as the mean of the next 5000 time steps"??

We have re-written a lot of the beginning of this section to better explain the motivation behind it. As we were designing the package for bias-adjustment for streamed data (another package not presented here), we came across this issue of converge of statistic summaries and feel that it is a useful discussion for others. In this section we hope to give users guidance in the other ways that summaries from one-pass algorithms can be used: such as in quantile-quantile mapping, and what should be considered in the process. This is now (we hope) much better explained in the introduction to this section.

16. L508: Thanks for the links. A few questions/suggestions: 1) might be good to list the GitHub links here as well? 2) is there a hosted version of the package docs somewhere? 3) how was the one-pass package used in making these figures? I didn't see any "import one pass" or "from one pass…" … looks like you reimplemented/"simulated" everything from scratch? If so, this means the package in the title of this manuscript wasn't used at all and

the whole manuscript is misleading. I also don't quite understand the numerics are returning round-off errors, where I would've expected exact answers (e.g., for the means)

Yes, we have included also the GitHub links along with the link to the hosted version of the package. The notebooks have now been reformatted using the package and we apologize for the confusion that they weren't originally included.

---

## Author Response (AR1)

**Response to Reviewers - 2025-28**

**I.  RESPONSE TO REVIEWER 1: REVIEWER SUMMARY**

In this manuscript, the authors present an alternative way to process "on-the-fly" (online) model output data, with the motivation being reduced data/memory usage and potential ease for users. Ultimately, this is an interesting manuscript, albeit weird. I think it can be made worthy of publication upon a revision. In general, the authors seem to be factual, careful, and nuanced. I think most of the work is relatively high quality. There are some problems though (the first one potentially fatal for the manuscript).

We would like to thank the reviewer for the time and effort to review our manuscript. We have made significant changes to the manuscript and have compiled a point-by-point response.

**II.  MAJOR COMMENTS:**

1. My reading of the manuscript is that it is making use of an algorithm package (whose version is in the title no less), but upon examination of the code availability section, I don't think the authors actually used the package at all in their analysis. Instead, they "simulated" using the package. I find this quite odd. I wrote a comment about this at the end. I don't know if this is intentional or not, but this is pretty deceptive. I invite the authors to explain, and I will keep an open mind.

   The reviewer raises a valid point here. To explain, the source code of the One_Pass package was not originally introduced due to contractual obligations. As we still felt that the analysis of the algorithm's performance was worthwhile, the backbone of the package was extracted so as to produce the same results, just not using the easy user interface and functionality of the One_Pass. This, however, was not deemed appropriate. When we were granted permission to make the package open source, we did so but did not update the corresponding notebooks, which was a mistake on our end and we thank the reviewer for spotting it. We have now updated the notebooks accordingly, to create the same Figures as with the old notebooks, but produced with the code of the One_Pass package, see `https://github.com/kat-grayson/one_pass_algorithms_paper?tab=readme-ov-file`. These notebooks now show the Figure created using the One_Pass configuration dictionary developed

with simple key:value pairs that allows the user to define their required statistic over the required time period.

We do note however that in some of the notebooks relating to the t-digest algorithm we also directly import the underlying Crick t-digest package [1] used in the One_Pass for analysis. This is because the One_Pass package acts as a user interface, hiding the background computation of the t-digest and will simply return the final percentile or histogram (depending on the user request). The One_Pass does not expose the inner workings of the t-digest to the user. In this manuscript analysis however, we also provide details on the cluster sizes and weights of the underlying t-digest objects. This information is not required when examining the final percentile, and is therefore not made available via the One_Pass package. However, as it interesting for the analysis and explanation of how the t-digest clusters work (e.g. Figure 3(a) in the manuscript) we obtain it directly from the t-digest package. Overall, we understand that the reviewer perceived this as a deception, but there was never an intent to give this impression.

2. The authors didn't explain how this type of algorithm/package would be used with an actual climate model running. Maybe it would be used in a futuristic cloud-compute setup? (They do cite some cloud-computing works.) I think this manuscript would greatly benefit from a detailed example (ideally a workflow) of how this would be implemented in an end-to-end fashion.

We would like to thank the reviewer for this comment and agree that it's current use was not made clear. We have included a section 'Design and implementation into a workflow' (Sect 2.) in the revised version of the manuscript. In this section, we describe the design choices of the package in the context of the project that it was built for. We explain the basic configuration and how it is currently used in an HPC workflow via a workflow manager. Originally this was not included due to limitations of not allowing the package to be open source, so we kept the scope of the paper to the more theoretical implementation. After clearing the contractual limitations that originally lead to the exclusion of the source code of the One_Pass package, we are happy to be able to include this section.

3. I think this manuscript would benefit from a comparison to the "online diagnostics" route. In the "online diagnostics" route, climate scientists (and developers) write functionalities to output specific items of interests, without needing too much data. Climate models already

do online mean, min, max, etc. in their output streams — nothing here is novel. That could easily be extended to all sorts of statistics. In fact, it could be extended in a composable fashion to even more complex and variable algorithms. For example, consider the following: "I want to calculate the globally averaged temperature at cloud-top but with a threshold of 280 K". An algorithm can be written to identify cloud top, then finding the temperature there, considering the threshold, and then doing a horizontal reduction in a composable way. This algorithm can then be run after each time step in the model, and storing in memory the variable to be output or writing it out immediately. I think the authors could improve this manuscript if they compare and contrast their one-pass streamed approach to that of models natively doing more online calculations themselves (potentially on idling CPUs, if the models run on GPUs) instead of an external module/package.

While it is true that models often provide mechanisms to compute variable statistics during simulation runs, the key here is that a model has no obligation to be tailored to the specific needs of a data consumer downstream. Here we define data consumers as external users who are not involved in the climate modelling process (i.e. not model developers) and traditionally access model output via static online archives. While it is common for models to produce online calculations of monthly means of certain variables, getting different modelling groups to align on specific statistics for downstream users (e.g. 99th percentile of wind speed) would be challenging, to say the least. In the new section 2 we describe the design choices behind the One_Pass package, and that it needs to be able to integrate with a variety of GCMs. We also outline the concept of the ClimateDT which operates with the core idea that data consumers may enter or leave the simulation data stream at any time, according to their needs, and do not enforce their needs onto the design of the model. Instead, with the One_Pass they are provided with a tool to select their fields of interest, as well as the aggregation method and frequency for those fields. The One_Pass performs the necessary statistical operations at a later stage downstream by an independent process, making the whole ecosystem more scaleable. We hope the inclusion of the new section 2 has given this more context and also addresses the minor comment 7 for for L35–42.

Minor comments

1. L2: nitpick: "as such" is kinda weird here in that it made me think running at high-res was a consequence of the preceding sentence, but it actually is more of a antecedent of the

following sentence.

We've removed this along with the first sentence in an attempt to condense and consolidate the abstract.

2. L5: nitpick: probably misplaced comma (I'd move it to after HPC instead)

   Fixed.

3. L7: the phrase "data streaming" is introduced like a known quantity that is specific climate science, but in fact, it is neither known nor usually associated with climate science. Maybe jargon? If I were you, I would consider removing "data stream;" (the phrase with the semicolon) and let the sentence stand without the disruption

   Data streaming is now properly defined in the abstract (lines 5-6) and in the introduction lines (38-39). It is a key concept that requires explaining as without projects adopting this method, there is no need for one-pass algorithms or the package. We have made sure to better define all jargon used in the manuscript (e.g. data consumers line 34).

4. L9: one-pass may necessitate explanation here (so I would rephrase)

   We've better defined these terms in the abstract (line 8-9 abstract) and introduction making sure not to introduce jargon without explanation.

5. L9: intelligent doesn't seem like the right word (maybe efficient?) — these techniques aren't learning things, or are there?

   You're right, they're not learning algorithms, we've removed this adjective in the rephrasing. Also removed in the introduction.

6. L17: I think "well within the acceptable bounds" is an underestimated. Imho and understanding, these algorithms basically recover accuracy to within an insurmountable precision limit, so basically as good as one would get anyway. I would rephrase to finish with a stronger point

   Agreed, we have rephrased to make this a stronger point.

7. L35–42: I think this may surprise an average climate scientist involved in model development and evaluation (such as myself). We always had the ability to access data before simulations are done and one could obviously write stuff in a continuous manner. I think what

you want to highlight is that you're algorithmically calculating some interesting statistics that capture specific interests. That is not really novel, not unique, but it is interesting and worthy of publication. In other words, I don't see the "novel" method and "unprecedented" reduction of "meaningful" output. Can you please (carefully) elaborate?

We understand the reviewers concern here and agree that the paradigm that the One_Pass package has been built around has not been fully explained. Indeed, someone involved in model development can write and access the climate model data before the whole simulation is finished, however model developers are not the target user of the One_Pass package. The package was designed as part of a project that aims to integrate climate modelling with impact modelling, through the use of data streaming. We use the term data consumer to refer to anyone who would traditionally access climate model data from static archives. The aim of the ClimateDT project is that these data consumers can become part of the ClimateDT ecosystem and access directly the climate model output. This means someone who would traditionally access data via an online archive could access the streamed data and start working with it without having to wait for the full model run to be finished and published online. Based on your comment, we have modified the introduction significantly to convey data consumers as being those who don't necessarily work in the model development and traditionally are limited with respect to the data being available to them. We have included the new section 2 (already detailed above) which puts the development of the One_Pass into the context of the project and hope this clarifies the motivation behind this work.

8. L70: the last sentence here gives the wrong impression of what you're trying to say. Your manuscript is supposed to showcase those algorithms, so "requesting" readers to go follow some other package documentation may not be the best way to politely say, "We don't discuss the code implementation of specific algorithmic details related to each statistic, and we instead focus on showing their utility to climate analysis" . . .

Thank you for this comment. We have more thoroughly embedded the differences between the One_Pass package and the aim of the paper, which as you say, is looking at the algorithms utility. We have rephrased L70 and tried to make it clear that examples of the package can be found online while here we discuss accuracy and provide guidance on their use.

9. L194: I'd elaborate on the different results part

We have expanded this statement to provide some more details. However when sampling other packages we did not find substantial differences between the implementations. Our decision was to use this particular implementation was also based on how often the package was being maintained.

10. L219: I think you can say that plainly at the outset without taking the reader on an unnecessary voyage. I am not sure if the added info (analysis) is informative otherwise. Feel free to disagree.

We realize this paragraph was quite convoluted. We have re-worded to introduce the main point at the start of the paragraph and have tried to condense to make the message clearer.

11. L268: so, the NumPy-calculated one is the "reference" right? (Note, I would refer to NumPy like they refer to it in their papers/docs, uppercase N and P)

Yes, NumPy is the reference. We've made this explicit in the introduction (line 66) that NumPy is always used for the 'conventional' method and fixed the casing.

12. Section 5.3: If I were you, I would simply delete this section. Your concluding paragraph (around L385) basically shows that most of the preceding analysis/"results" should be taken with a giant grain of salt. I would simply avoid the distraction, maybe add something about how the analysis in 5.2 could done on precipitation with a meaningful cutoff and/or a different type of underlying assumption (distribution type)

We don't agree that removing this section is a good idea. The aim of the paper is to show how well the package will represent different statistics for different climate variables that are of interest to the climate community. Accurately capturing extreme precipitation is essential and we want to provide a detailed analysis of how the package performs on this front. We have however substantially revised this section, reducing the text to make it more concise and updating the Figure. Now we show comparisons between the histograms for the t-digest method and the NumPy method which aid in the explanation of the differences for the 99th percentile estimate.

13. L431: But this doesn't make sense to me. As a scientist, what I care about is the stability of the value, not an arbitrarily defined metric of convergence. In Figure 6, $\sigma_n$ goes from roughly 0.35 at 200 samples to 0.15 at 600 samples, but it is within those "convergence" lines.

We use the convergence rate `https://en.wikipedia.org/wiki/Rate_of_convergence` to define how long it takes for the underlying statistic summary to be representative of the statistic of whole data set. We are not claiming the statistic has reached a final or static value. Climate variables will exhibit temporal variability and long-term trends, and we do not assume that their statistical properties remain fixed over time. Rather, our focus is on when the variability in the estimate of $\sigma_n$ itself has sufficiently diminished — i.e., when its fluctuations due to sample size limitations become negligible compared to inherent data variability. This is to allow the statistic summaries to be used in methods such as quantile-quantile mapping for bias-adjustment of streamed data. We have modified the introduction of the section considerably to clarify this.

14. L432: Ok, but how does that impact your assertions earlier about all the savings and not having to wait for model simulations to run a lot of steps? I guess you're saying, one kinda has to wait a lot of steps to get "accurate" stuff out of these one-pass algorithms? Like 200 steps or so?

The analysis conducted here doesn't impact earlier assertions about the use of one-pass methods. As in previous sections, here we are benefiting from the memory reduction of the one-pass algorithms and the ability to work with the data stream. Our point about timely access was meant to compare, for example, those data consumers with impact models who currently access data from online, static archives. In the overall aim of the ClimateDT, these types of impact models will be able to embed into the ClimateDT workflow and access directly the model output via this streaming paradigm. We hope that with the inclusion of the new section this has been made much clearer. If you don't agree, we would greatly appreciate more thoughts on this.

15. Section 6: Like Section 5.3, I think the convergence analysis is likely misleading, potentially counterproductive, and perhaps better left out. Your goal in this manuscript is to showcase something useful for users of this method. My assumption is that if people choose to use this type of one-pass algorithm, they would do so on relatively high-frequency data and they would understand the risk of under-sampling. Maybe I am just not getting what you're trying to do here? Could you motivate it better? Is it anything other than something like "don't take the mean of the first 5 time steps as the mean of the next 5000 time steps"??

We understand the concern of the reviewer and we have re-written a lot of the beginning of

this section to better explain the motivation behind it (See lines 413 - 440). As the output of this package is used as input for bias-adjustment of streamed data, we came across this issue of converge of statistic summaries and feel that it is a useful discussion for others. We have also motivated this topic in the introduction (See lines 45).

16. L508: Thanks for the links. A few questions/suggestions: 1) might be good to list the GitHub links here as well? 2) is there a hosted version of the package docs somewhere? 3) how was the one-pass package used in making these figures? I didn't see any "import one pass" or "from one pass..." ... looks like you reimplemented/"simulated" everything from scratch? If so, this means the package in the title of this manuscript wasn't used at all and the whole manuscript is misleading. I also don't quite understand the numerics are returning round-off errors, where I would've expected exact answers (e.g., for the means)

Yes, we have included also the GitHub links along with the link to the hosted version of the package. The notebooks have now been reformatted using the package and we apologize for the confusion that they weren't originally included.

**III.  RESPONSE TO REVIEWER 2: REVIEWER SUMMARY**

This article describes techniques for on-line statistical analysis of climate model output which can reduce the amount of output data without needing to degrade the spatial resolution of the output. In particular a sophisticated, novel t-digest algorithm is described for on-line generation of percentiles and histograms. The publication is timely given the increasingly large volumes of output being created by increasingly complex and high-resolution numerical models, especially for the emerging class of km-scale models many modeling centers are developing. In particular the quantification of memory savings is a powerful selling point. This paper should be published and GMD is the ideal journal for this purpose. However I was somewhat confused by the discussion of the t-digest method and would like to have some more clarification on a number of points.

We would like to thank the reviewer for their effort in reviewing our manuscript. We appreciate the very detail comments regarding the t-digest method and provide a point-by-point reply below.

**IV.  MAJOR COMMENTS:**

1. I am unfamiliar with some of the terminology. Does the "scale function" k(q) introduced in equation (6) define the "edge values" for each of the bins/clusters? If so, an illustration (showing the domains for each cluster as a function of delta) would be useful.

    Thank you for the comments relating to the t-digest description. We have taken them all into account and have significantly modified the introduction of this section (see lines 221 - 240). The scale-function is a monotonically increasing function and the location of each cluster's mean is defined by it's slope (k-space). Due to it's hyperbolic shape, $k$ has a steeper gradient at the tails, resulting in smaller cluster sizes over these ends. We have now shown this visually in a new Figure 3(a), which shows the shape of the scale function over a range different compression parameters. Note a similar figure is shown in Dunning, T. and Ertl, O.: Computing Extremely Accurate Quantiles Using t-Digests, arXiv[preprint],arXiv:1902.04023, 2019, Figure 1. In the new figure the clusters are shown for $\delta = 20$ and $\delta = 100$ for two different t-digests that represent the same data set $X_{500}$. We have used the size of the gray dots to indicate the weight of the cluster, showing that fewer clusters with larger weights are used for $\delta = 20$, than 100. Exactly how the edges of clusters are defined depends on the size of the cluster in $k$ space. This is described in detail in Dunning and Ertl [2] and we refer the

reader there for technical details of this process. We have included a short description in the paper avoiding detailed methodology as we want to focus on the accuracy of the algorithms implementation for different climate variables.

2. Also, should the same scale function be used for each variable? While having a somewhat equal-spaced set of bins is possibly a good choice for wind speed, this would not be as useful for precipitation since (as is shown in Section 5.3) small values are far more common than large ones, and so the small bin will fill up very quickly.

The reviewer raises an excellent point here and we agree that for precipitation a different scale function would be preferable. The aim of this work is to show the performance of the One_Pass package (v0.8.0) (upgraded version since previous submission) in representing these variables so we have performed the analysis using the scale function currently implemented. We comment at the end of section 6.3 (previously 5.3) that these results would definitely be improved by using a scale function specifically designed for precipitation, however this is outside the scope of the paper. We also note that one easy way to bypass this problem would be to include a cut-off value for the precipitation, so simply ignoring all values below say 0.1 mm/day to make best use of the available clusters. We have modified this section to make the text more concise and included histograms obtained with NumPy in the Figure to show more visually the differences between the one-pass and the conventional method.

3. Also, how the t-digest method goes from its clusters to creating percentiles and histograms is unclear. Could a more concrete demonstration of this be shown? Also, are the numpy percentiles and histograms created by a similar algorithm, or are these the exact percentiles and histograms computed from all of the data without approximation?

We have addressed both these comments in the new Sec 6.1. Specifically we have added the sentence "These clusters are ultimately converted to a percentile or a histogram (where the number of samples in each bin may have non-integer values due the underlying cluster representation), based on the closest cluster mean to the required percentile. " We hope that this, along with the new Figure which shows the cluster weights in relation to their quantile will make this conversion clearer. For the NumPy implementation, it does indeed have access to the full data without approximation. We have added a clear statement explaining this in the last paragraph of Sec 6.1.

4. I was very glad to see the careful validation of the simpler one-pass algorithms (mean, standard deviation) and the striking demonstration that the errors are at rounding level. I had more trouble understanding Figures 4 and 5. It is not immediately clear that the red and orange dots represent the different locations in panels (b)–(e) without a legend; and the histograms inset in panels (b) and (e) are so small as to be almost impossible to interpret. It would be more useful to overplot these on a larger panel to more directly compare the two. I also don't quite understand the main result in panels (b) and (e): the shaded area is the typical wind turbine operating tolerance for these locations, and each dot is another percentile larger (first dot is 1st percentile, last is 99th)? Overall I think splitting out the panels and/or insets into different figures, providing better labels and annotations, and enlarging some plots would make it much easier to read and understand.

Thank you for this comment. We have significantly updated both Figure 4 and 5. They both now contain legends to indicate that the range of colors signify particular locations. We agree this was not so clear before. We have also converted the histogram inserts in Figure 4 into their own panels to allow for a proper comparison between the t-digest and NumPy methods. This has further been done in Figure 5 to allow for continuity between the Figures and also show how the full precipitation distribution is represented. For all histograms, their colors now correspond to the location they represent, marked in the legend.

5. The discussion of convergence of climatologies in Section 6 is interesting and does get at the vital scientific question of how much data is needed to create a useful climatology. However, given that the one-pass algorithms create nearly-exact statistics (for means and standard deviations) or up to the cluster's sampling uncertainty (t-digest) for the model run segment being considered; constructing a climatology for bias correction or other purposes would be a downstream step that would presumably mean aggregating statistics over a large number of segments and/or individual simulations. Would not this analysis then say how long the simulation needs to be to create an accurate climatology, and not have much effect on the algorithms of this paper?

We have changed the introduction to this section to better explain motivation as it was not clear before. Indeed, this section boils down to "how much data do you need for a climatology/summary". However, in this context it falls under the one-pass umbrella as we consider the bias-adjustment of streamed climate data. Bias adjustment is one example here for a data

consumer that needs to process the data immediately when it is produced, hence it is required to know when the summary statistic can be used. What we are aiming to do with this section is to show other applications for the one-pass algorithms and how the rolling summary of any statistic $S_n$ can be considered a reliable estimate of the statistic of the total distribution after a certain number of samples. This is crucial when designing bias-adjustment of streamed data or other downstream applications. We hope we have addressed the reviewers comment and explained how this is another 'application' of the one-pass algorithms.

Minor comments

1. There is precedent for on-line calculations of statistics such as means, extrema, standard deviations and so on. See the "data reduction" techniques here, for instance: `https://github.com/NOAA-GFDL/FMS/blob/main/diag_manager/diag_yaml_format.md` and `https://github.com/NOAA-GFDL/FMS/blob/main/docs/diag_table.md`

   We have now included a new section (Sect. 2) to discuss the context of the development of this package and why it was necessary to decouple it from online calculations produced by the models. We have used this repo as as a reference for those online calculations.

2. - Line 171: Should "extremely values" be "extremely small values"?

   Fixed.
* * *
[1] Crist-Harif, J.: `https://github.com/dask/crick`, 2023. 2

[2] Dunning, T. and Ertl, O.: Computing Extremely Accurate Quantiles Using t-Digests, `arXiv[preprint],arXiv:1902.04023`, 2019. 9